# Prognostics for pain in osteoarthritis: Do clinical measures predict pain after total joint replacement?

Joana Barroso[1,2,3], Kenta Wakaizumi[2,4], Diane Reckziegel[3], João Pinto-Ramos[5], Thomas Schnitzer[2,6], Vasco Galhardo[1], A. Vania Apkarian[2,3]*

1 Departamento de Biomedicina, Faculdade de Medicina, Universidade do Porto, Porto, Portugal; Instituto de Investigação e Inovação em Saúde—i3S, Universidade do Porto, Porto, Portugal, 2 Department of Physical Medicine and Rehabilitation, Northwestern University, Feinberg School of Medicine, Chicago, IL, United States of America, 3 Department of Physiology, Northwestern University, Feinberg School of Medicine, Chicago, IL, United States of America, 4 Shirley Ryan AbilityLab, Chicago, IL, United States of America, 5 Departamento de Medicina Física e de Reabilitação, Centro Hospitalar e Universitário de São João, Porto, Portugal, 6 Department of Rheumatology, Northwestern University, Feinberg School of Medicine, Chicago, IL, United States of America

* a-apkarian@northwestern.edu

**Data Availability Statement:** All relevant data are within the manuscript.

**Funding:** J.B. was funded through CCDRN [Norte-08-5369-FSE-000026], OARSI Collaborative

## Abstract

A significant proportion of osteoarthritis (OA) patients continue to experience moderate to severe pain after total joint replacement (TJR). Preoperative factors related to pain persistence are mainly studied using individual predictor variables and distinct pain outcomes, thus leading to a lack of consensus regarding the influence of preoperative parameters on post-TJR pain. In this prospective observational study, we evaluated knee and hip OA patients before, 3 and 6 months post-TJR searching for clinical predictors of pain persistence. We assessed multiple measures of quality, mood, affect, health and quality of life, together with radiographic evaluation and performance-based tasks, modeling four distinct pain outcomes. Multivariate regression models and network analysis were applied to pain related biopsychosocial measures and their changes with surgery. A total of 106 patients completed the study. Pre-surgical pain levels were not related to post-surgical residual pain. Although distinct pain scales were associated with different aspects of post-surgical pain, multi-factorial models did not reliably predict post-surgical pain in knee OA (across four distinct pain scales) and did not generalize to hip OA. However, network analysis showed significant changes in biopsychosocial-defined OA personality post-surgery, in both groups. Our results show that although tested clinical and biopsychosocial variables reorganize after TJR in OA, their presurgical values are not predictive of post-surgery pain. Derivation of prognostic markers for pain persistence after TJR will require more comprehensive understanding of underlying mechanisms.

## Introduction

Osteoarthritis is the most common cause of arthritis worldwide and a major source of chronic musculoskeletal pain. Although nociceptive inputs elicited by joint degeneration and chronic

Scholarship 2018 and Luso-American Development Foundation R&D@PhD scholarship grant. This research did not receive other specific funding from agencies in the public or commercial sectors.

**Competing interests:** The authors have declared that no competing interests exist.

inflammation are commonly recognized as contributing factors, current understanding of OA pain pathophysiology remains incomplete. In the last few years, a growing body of research indicates that altered peripheral and central nociceptive processes are influential [1]. This is substantiated by the discordance in joint structural damage and pain intensity [2], but also by the results of surgical treatment [3]. Total joint replacement (TJR) is an effective and safe intervention for advanced hip and knee OA; nevertheless, an important proportion of patients still report moderate to severe persistent pain post-TJR, not attributable to identifiable surgical or clinical complications. In the case of knee OA (KOA), persistent post-surgical pain is reported in about 20% of the patients. For hip OA (HOA), this number appears to be lower (up to 10%) [4].

Persistent post-TJR pain remains minimally understood. Post-surgical pain is defined as pain that occurs or intensifies after the procedure and lasts for at least 3 months [5]. However, in OA, long- lasting chronic pain pre-exists and is the main impetus for undergoing TJR, which complicates understanding post-surgical outcomes. Thus, it remains unclear the extent to which the post-surgical OA pain reflects residual presurgical pain, surgery induced pain, or some complex combination of both [6].

Regarding risk factors for pain persistence after TJR, those have been proposed, are mainly for KOA [7]. Pain intensity prior to surgery, disproportion between pain intensity and articular damage, neuropathic-like symptoms, psychosocial factors such as pain catastrophizing and poor coping strategies are commonly referenced as important predictive factors [8] [9] [10] [11]. Although these have been studied repeatedly, there is extensive variation of outcome measures used and there is no agreement on which measures are optimal to assess chronic pain after TJR [7]. The proposed risk factors across studies are often diverse, tested through univariate associations, based on different study designs and analysis methods, thus the quality of evidence on prognostic factors for recovery after total knee replacement (TKA) remains low [12].

Here, in a prospective cohort study, we test the hypothesis that presurgical pain and pain-related psychosocial parameters contribute to post-TJR pain in knee and hip OA. Our main aims are: 1) Test if baseline pain ratings relate to post-surgery pain levels; 2) examine how distinct pain measurement instruments relate to different clinical and biopsychological aspects of OA pain; 3) develop and evaluate models predictive of pain and pain relief after TJR; 4) use network analysis to assess the reorganization of pain related clinical and biopsychosocial properties of the personality of KOA and HOA patients after TJR.

## Materials and methods

### Study sample

KOA and HOA patients with clinical indications for primary arthroplasty surgery participated in this longitudinal observational study. The present report is part of a brain neuroimaging study, studying central mechanisms in osteoarthritis, which will be reported subsequently.

Enrollment took place at the Orthopedic Surgery Department of *Centro Hospitalar de São João*, a tertiary care hospital in Porto, Portugal. Study protocol was approved by the local Ethics Committee–*Comissão de Ética para a Saúde, Centro Hospitalar de São João*, and all participants provided informed written consent prior to partaking in the study. Sample size was determined by the number of patients waiting for surgery who met the eligibility criteria for the study, during a period of 20 months. Initial evaluation happened 1–3 months before TJR surgery and follow-up continued up to 6 months after surgery. A total of 95 knee OA and 25 hip OA patients, and 37 healthy control subjects were included (the last group not studied in this report).

Eligible patients met the following inclusion criteria: age between 45 and 75 years-old; diagnosis of HOA and KOA according to the clinical classification criteria of the *American College of Rheumatology*, and surgical indications for TJR (criteria for surgery selection was moderate to severe pain and quality of life impairment, after clinical and radiological evaluation and medical decision by a certified orthopedic surgeon in our center). Patients were excluded when there was evidence of secondary OA due to congenital or development diseases and inflammatory bone and articular diseases. Bilateral OA with predicted indication for contralateral arthroplasty in the following year, other chronic pain conditions (e.g., fibromyalgia; chronic pelvic pain) and chronic neurological or psychiatric disease (e.g., depression major, dementia, obsessive compulsive disorders, Parkinson's disease, demyelinating diseases, peripheral sensory neuropathy), were also exclusion criteria, as well as cognitive impairment. Previous history of stroke or traumatic brain injury was also exclusionary. Secondary OA following history of minor trauma or previous arthroscopy surgery due to ligamentous/meniscal injury was not an exclusion criterion.

## Study design

This study comprised a total of 4 visits. Patients were initially assessed 1–3 months before surgery (V1). A second pre-surgical visit was held 2 to 6 weeks prior surgery (V2). Two post-surgical visits (V3-V4) occurred at 3 months and 6 months post-surgery. Specific data collected at each visit are shown in **Fig 1**. During visits 1, 3 and 4 patients were assessed for: (1) Clinical and socio-demographic properties; (2) physical function–performance-based tests; (3) radiographic evaluation, (4) pain, mood and health questionnaires; brain imaging was performed at visits 2 and 4.

## Measures

**Clinical and demographic assessment.**  Demographic profiling, acquired at V1, included age, education and professional status. Medical data concerning height and weight, pre-surgical co-morbid conditions, previous surgeries, general medication and smoking habits were recorded at patient interview and by clinical charts analysis. A clinical questionnaire regarding the history and evolution of knee pain assessed pain onset, duration and frequency; pain medication and previous non-pharmacological treatments. The Medicine Quantification Scale (MQS) was used to score type and dose of pain medication [13]

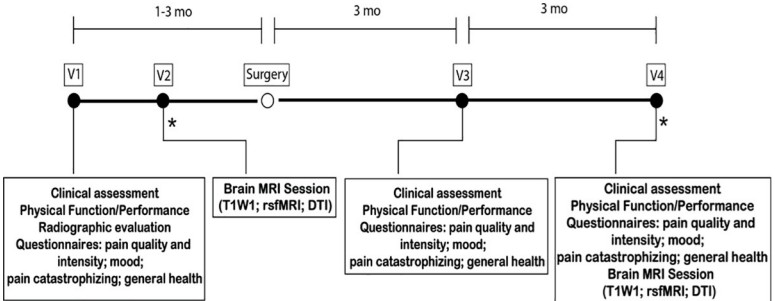

**Fig 1. Experimental design, timeline and data collected.** Knee and hip osteoarthritis patients entered a 4 visit (V1-V4), pre- and post-total joint replacement surgery, longitudinal, observational study. V1 and V2 occurred before surgery. V3 and V4 took place at 3 and 6 months after surgery. At each visit, participants underwent a series of assessments. *Brain MRI session. mo, months; MRI, magnetic resonance imaging; T1WI, T1-weighted imaging; rsfMRI, resting state functional magnetic resonance imaging; DTI, diffusion tensor imaging.

At the post-surgical visits (V3-V4) a second clinical questionnaire assessing pain recovery, time to recovery, patient satisfaction, pain medication, use of health care services and rehabilitation protocol was administered.

**Physical function–performance-based tests.** Physical function was assessed with two different tests, depending on the activity measured. Ambulatory transitions were evaluated with the Timed up and go test (TUG) [14], and aerobic capacity/walking long distances with the six-minute walk test (6MWT) [15–17]. These tests were selected based on the *OARSI* 2013 recommendations [18].

**Radiographic assessment.** As part of standard hospital protocol, patients scheduled for TJR had bilateral joint radiographs during the 6 months before surgery. Knee OA radiographs were taken in two views: anterior-posterior (AP) weight-bearing with knee flexion at 20˚ and foot internal rotation at 5˚, and horizontal beam lateral view, with lateromedial projection, the patient in supine position and the knee flexed at 30˚. Hip OA patients had AP supine radiograph of the pelvis, with lower limbs internally rotated 15˚ degrees from the hip.

Radiographs were scored accordingly to the Kellgreen-Lawrence (KL) classification—grades 0 to 4 [19], by two trained radiologists. The first classified the whole sample, the second classified half of the subjects for inter-reliability measurement. Both researchers were blind to the clinical data of the patients when scoring. Inter-rater reliability was determined for KOA imaging only and the intra-class correlation coefficient of KL grading was 0.91 (95% confidence interval 0.80–0.93).

**Questionnaires–Pain, mood and health.** Seven questionnaires were administered by a trained clinician, during face-to-face interview. They were administered both before surgery (V1), and in the post-surgical visits (V3-V4). The repeated use of the same measures allowed us to track changes concerning intensity and quality of pain, emotion and affect, health and quality of life. All questionnaires were used in their Portuguese version, and validation data regarding their adequate context validity, internal consistency and test-retest reliability were consulted and hereby cited. We assessed: 1) KOOS, HOOS, validated injury and OA outcome scores for knee and hip [20–22]; 2) Brief Pain Inventory–Short Form (BPI) [23–25]; 3) McGill Pain Questionnaire (MPQ) [26, 27]; 4) Doleur Neuropathique en 4 Questions (DN4) [25, 28]; 5) Hospital Anxiety and Depression Scale (HADS) [29, 30]; 6) Pain Catastrophizing Scale (PCS) [25, 31]; and 7) SF36-item Short Form Survey (SF36) [32, 33].

**Primary outcome variables.** Primary outcome variables were part of the questionnaires/clinical assessments and consisted of 4 distinct pain intensity related scales/subscales: Numeric Rate Scale (NRS); BPI–Pain Severity; KOOS Pain and HOOS Pain, as clinically appropriate; SF36 Bodily Pain, here addressed specifically for knee/hip articular pain.

For each of the 4 outcome measures and for an aggregate of all four, we examined relationships for pain relief post-surgery on a per subject basis, by calculating residual pain: %residual pain = 100 –(100 *(average pain pre-surgery—post-surgery pain [at 3, or 6, months])/ average pain pre-surgery)). Thus, 100% residual pain = no change in a given pain measure between before and after surgery; 0% residual pain = complete relief from initial pain; while values >100% indicate worsening of pain post-surgery.

As the literature more commonly reports on the effect of pre-surgery baseline pain [9], we also examined and modeled influence of baseline pain on post-TJR pain.

## Statistical analysis

All data from the reported measures were manually entered by the same researcher. Regarding missing data, if 30% or more was missing from a questionnaire (total or sub-score if

applicable), it should be excluded. If missing data were less than the threshold 30%, we used the mean of the total score/sub-score to fill in missing items.

Descriptive statistics were used to describe the study sample, with continuous variables presented as mean and standard deviations and categorical data as numbers and percentages. Comparisons between the two OA groups used independent sample t-tests or Chi-square($X^2$) tests, for continuous parametrical variables and categorical data respectively.

Interrelationship of the primary outcome variables (all scored on a 0–10 score) was assessed through correlation analysis using Pearson product-moment tests. Fischer's z tests were used to evaluate differences between correlation coefficients at baseline, 3 and 6 months. The effects of time (pre-, 3- and 6-months post-surgery), type of OA and pain outcome measure on pain intensity were studied with a three-way mixed ANOVA. Following the initial procedure, two-way interactions and simple main effects were considered and pairwise comparisons with Bonferroni adjustments were performed.

Due to the high number of clinical and psychological measures collected, a data dimensionality reduction from all 19 subscales of 7 questionnaires and 2 physical performance scores was achieved using a principal component analysis (PCA) in KOA patients at baseline. This allows to reduce the data into fewer dimensions, while retaining underlying trends and patterns. Overall and individual Kaiser-Meyer-Olkin measures were 0.86 and >0.5 respectively. Threshold for component retention was set on eigenvalues >1.0, together with visual inspection of the scree plot for evaluation of the inflection point. A factor rotation on the obtained components was applied using a Promax oblique rotation technique. Threshold of factor loading was set on 0.5/-0.5 and components were labeled given the observed loadings. Due to the limited number of subjects available in the HOA group, we generated the component values using the same weights retrieved with PCA for the KOA, which enables direct comparison of TJR effects on network properties.

Different regression analysis techniques were used to model pain outcomes in KOA and HOA. For KOA, multifactorial regression models were generated using a stepwise forward and backward selection method, in an automatic step-by-step iterative construction of the model. Significance level to enter (α-to-enter) was set at 0.05 and α-to-remove at 0.10. To test if the models obtained in KOA replicated in HOA patients, and due to a smaller sample size in this group, we applied a multiple linear regression analysis in HOA, entering as independent variables the predictor factors uncovered for KOA, thus testing the extent of shared factors between the two conditions. For all regression models, assumptions of linearity, independence of observations, homoscedasticity and absence of multicollinearity were met, and residuals were approximately normally distributed in all models. Outliers were detected by examining studentized deleted residuals, any values greater than ± 3 standard deviations were removed. Throughout all models, no more than 3 cases were removed.

Correlation matrices of the clinical and psychological variables (questionnaires subscales and physical performance scores) were represented as binarized networks, constructed at the 25% stronger correlations for each matrix (KOA/HOA at baseline, 3- and 6-months post-surgery), and visualized using the software Cystoscape (v3.6.1, http://www.cytoscape.org). For each network, questionnaire measures were represented as nodes and the thresholded correlations as edges. Network communities were derived from the previous PCA. Two network graph measures were computed to characterize and quantify topological changes, using the Matlab Brain Connectivity Toolbox [34]. Clustering coefficient is a measure of the extent to which nodes in a graph tend to cluster together. Nodes have the trend to create groups characterized by a high density of connections. We computed local clustering coefficient of all nodes, and averaged them, reflecting the overall level of clustering in a network, from 0 (no clustering) to 1 (maximal clustering). The second calculated measure, modularity, refers to the

compartmentalization and interrelation of modules in a network. Modules can be defined as sets of nodes densely connected among themselves and poorly connected to other regions of the network. Using the Louvain community detection algorithm, averaged over 100 computed repetitions, we obtained values that vary from 0 (random network) and 1 (highly structured network).

We studied the changes in the strength of connectivity for all networks, calculating the change in correlation coefficients for all pairs of subscales from baseline to three and six months, and averaging these over the entire networks, obtaining the mean ΔR. For all inter and intra-group comparisons, regarding network measures and change in correlation coefficients, statistical probability was computed with 10,000 repeated random resampling.

All data were analyzed using the Statistical Package for the Social Sciences (IBM Corp. Released 2016. IBM SPSS Statistics for Windows, Version 24.0. Armonk, NY: IBM Corp), JMP software (JMP®, Version *14*. SAS Institute Inc., Cary, NC, 1989–2007) and MATLAB (MATLAB and Brain Connectivity Toolbox release 2016a, The Mathworks, Inc., Natick, Massachusetts, US).

## Results

### Recruitment, assessment and participant characteristics

A total of 94 KOA and 25 HOA patients were eligible and agreed to participate in this longitudinal, observational study. At 6 months, a total of 84 KOA and 22 HOA completed the study and were included in the analysis. **Fig 2** presents patient and control participants flowchart and timeline. Causes for withdrawal included: revision arthroplasty due to periprosthetic infection or prosthesis displacement (n = 4); other co-morbidities (n = 2, concomitant oncological disease) and voluntary withdrawal (n = 12). Most included patients had a complete dataset, except 8% (n = 8) had missing data for at least one questionnaire and for those no more than 12% of each questionnaire was missing. The missing data were imputed using the mean for each scale/subscale.

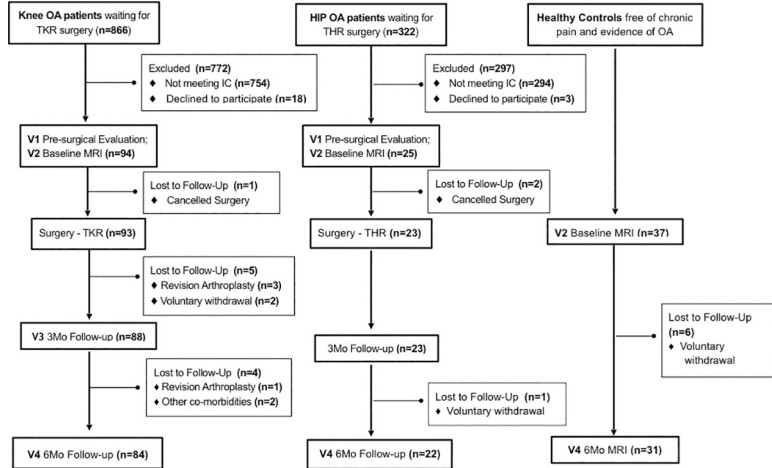

**Fig 2. Recruitment and retention for KOA and HOA, and healthy control participants.** The full battery of assessments was performed in osteoarthritis patients. Healthy individuals were recruited to act as controls in brain imaging analyses (not reported here). All patients were recruited from the same tertiary care hospital. Healthy participants were recruited from the general population in the Porto area. IC, Inclusion criteria; MRI, magnetic resonance imaging; KOA, knee osteoarthritis; HOA, hip osteoarthritis; THR, total hip replacement; TKR, total knee replacement.

**Table 1. Demographic characteristics of KOA and HOA patients.**

|  | KOA(n = 84) | HOA(n = 22) | t/X2 | p |
|---|---|---|---|---|
| Age, Mean±SD | 65.6 ± 0.7 | 60 ± 1.6 | 4.005 | **<0.001** |
| Gender, Count, % | 75 (79.8) / 19 (20.2) | 8 (32) / 17 (68) | 21.37 | **<0.001** |
| BMI, kg/m2, Mean±SD | 30.3 ± 0.5 | 28.2 ± 0.7 | 2.027 | **0.045** |
| Smoking Status, Count, % |  |  |  |  |
| Active smoker | 8 (8.5) | 2 (8) |  |  |
| Former smoker | 14 (14.9) | 6 (24) | 1.176 | 0.556 |
| Non-smoker | 72 (76.6) | 17 (68) |  |  |
| Education Level, Count, (%) |  |  |  |  |
| Primary school | 74 (78.4) | 17 (68) |  |  |
| Secondary school | 15 (16) | 5 (20) | 1.176 | 0.407 |
| Graduate | 5 (5.3) | 3 (12) |  |  |
| Professional Status, Count, (%) |  |  |  |  |
| Retired | 67 (70.5) | 9 (36) |  |  |
| Active | 16 (16.8) | 2 (8) | 14.424 | **0.006** |
| Medical leave | 12 (12.6) | 14 (56) |  |  |
| Habitation, Count, (%) |  |  |  |  |
| Alone | 17 (18.1) | 3 (12) | 0.523 | 0.48 |
| Cohabitation | 77 (81.9) | 22 (88) |  |  |

A total of 84 KOA and 22 HOA completed the study and were included in the analyses. Differences between groups were tested using T-test for continuous and parametric variables ($t$), and Chi-square tests for categorical data ($X^2$). P-values <0.05 were considered significant (bolded). BMI, body mass index; F/M = female/male; KOA, knee osteoarthritis; HOA, hip osteoarthritis.

**Table 1** describes HOA and KOA patients' demographic characteristics. Mean age of KOA patients was greater than that of HOA patients; the KOA group was predominantly female while the HOA group included mostly males. Body mass index (BMI) was higher in KOA than HOA patients. Smoking habits, educational level and habitation status were similar between HOA and KOA. Regarding occupational status, for the KOA group the most common status was retirement; HOA patients were mainly on medical leave, which relates to their differences in age.

## Pain intensity as a function of type of pain measurement instrument, surgery, time, and OA joint involvement

We examined the pain intensity determined by our four pain outcome measures (NRS, KOOS pain, BPI pain severity, SF-36 pain), both at baseline and after surgery, in KOA and HOA patients, and then evaluated their interrelationship (**Table 2**). All pain magnitudes decreased post-surgery, correlations among measures generally strengthened. Mean post-surgical pain levels (across all measures) was lower in the HOA group than in the KOA group, and the pain intensity estimate was highest with the SF-36 pain scale.

A three-way ANOVA was conducted to determine the effects of time (pre-, 3, 6, months post-surgery), the four pain outcome measures, and the type of joint OA, on pain intensity. We found a non-significant three-way interaction between these variables. The two-way interactions were statistically significant between pain measures and time ($F_{(6,624)} = 5.231$, $p<0.001$); type of OA and time ($F_{(2,624)} = 4.096$, $p = 0.022$); but not type of OA and types of pain measures ($F_{(3,624)} = 2.021$, $p = 0.129$). These results reveal that questionnaires show a similar rating pattern for hip and knee OA, but they vary in different ways over time.

**Table 2. Pain in KOA and HOA patients pre- and post-surgery, characterized with four pain outcome measures.**

| Pain Outcome Measure | | | | | |
|---|---|---|---|---|---|
| **Baseline** | | | | | |
| Knee OA (n = 84) | **M** | **SD** | **(1)** | **(2)** | **(3)** |
| (1) BPI Pain Severity | 4.79 | 1.5 | | | |
| (2) NRS | 6.53 | 1.67 | .792** | | |
| (3) KOOS Pain | 6.49 | 1.49 | .266** | .313** | |
| (4) SF-36 Pain | 7.04 | 1.82 | .162* | .288* | .475** |
| Hip OA (n = 22) | | | | | |
| (1) BPI Pain Severity | 4.38 | 1.52 | | | |
| (2) NRS | 6.09 | 1.66 | .801** | | |
| (3) HOOS Pain | 5.86 | 1.63 | .564* | .405 | |
| (4) SF-36 Pain | 6.49 | 1.55 | .547* | .474* | .631* |
| **3 Months** | | | | | |
| Knee OA (n = 84) | **M** | **SD** | **(1)** | **(2)** | **(3)** |
| (1) BPI Pain Severity | 1.69 | 1.48 | | | |
| (2) NRS | 1.89 | 2.03 | .922**† | | |
| (3) KOOS Pain | 2.45 | 1.96 | .837**† | .809**† | |
| (4) SF-36 Pain | 3.39 | 2.25 | .750**† | .771**† | .774**† |
| Hip OA (n = 22) | | | | | |
| (1) BPI Pain Severity | 0.54 | 1.05 | | | |
| (2) NRS | 0.55 | 1.06 | .920** | | |
| (3) HOOS Pain | 0.79 | 0.88 | .257 | .159 | |
| (4) SF-36 Pain | 1.95 | 1.71 | .329 | .356* | .381* |
| **6 Months** | | | | | |
| Knee OA (n = 84) | **M** | **SD** | **(1)** | **(2)** | **(3)** |
| (1) BPI Pain Severity | 1.70 | 1.53 | | | |
| (2) NRS | 2.01 | 1.9 | .930**† | | |
| (3) KOOS Pain | 2.19 | 2.17 | .813**† | .784**† | |
| (4) SF-36 Pain | 2.98 | 2.26 | .671*† | .652**† | .791**† |
| Hip OA (n = 22) | | | | | |
| (1) BPI Pain Severity | 0.63 | 0.76 | | | |
| (2) NRS | 0.73 | 0.93 | .961**† | | |
| (3) HOOS Pain | 0.80 | 0.77 | .379 | .280* | |
| (4) SF-36 Pain | 1.98 | 1.87 | .471* | .479* | .559** |

Four scales were used (BPI Pain Severity, NRS, HOOS Pain, SF-36 Pain; all presented on a 0–10 scale) to assess pain pre-surgery (Baseline) and 3, and 6 months post-surgery. Mean (M), standard deviation (SD) and Pearson's product-moment correlations between the 4 scales are presented. For both OA groups the four measures decrease in amplitude after surgery and are correlated with each other, improving following surgery in the KOA group.

* $p < 0.05$

** $p < 0.01$.

† Significant increase in correlation from baseline, $p < 0.05$.

KOA, knee osteoarthritis; HOA, hip osteoarthritis; BPI Severity, Brief Pain Inventory Pain: severity subscale; HOOS Pain, Hip Injury and Osteoarthritis Outcome Score: pain subscale; NRS, Numeric Rating Scale; SF36 Pain, Short-form (36) Health Survey: pain subscale.

Moreover, decrease in pain over time is larger for HOA in comparison to KOA. These results are further characterized in S1 Fig.

Main effects of pain measurement types were statistically significant at baseline, 3 and 6 months (F (3,315) = 66.6, p<0.001; F (3,315) = 51.03, p<0.001; F (3,315) = 41.02, p<0.001). Pairwise comparisons revealed that at baseline, pain intensity estimates were lowest for BPI

pain severity, (mean differences—NRS: -1.72 [-2.44, -0.99], KOOS/HOOS: -1.436 [-2.158, -0.71], SF36: -2.06 [-2.78, -1.4], p<0.001). At 3 and 6 months after surgery pain intensity was higher when measured by SF-36 pain (mean differences at 3 months: NRS: 1.269 [0.509,2.03], BPI:1.566 [0.8,2.33], KOOS [0.24,1.76], p<0.003; at 6 months: NRS: 1.038 [0.23,1,81], BPI: 1.354 [0.57,2.14], p<0.003). Thus, one cannot assume that these different measurements are equivalent.

Joint involvement was also significant: KOA patients had higher levels of reported pain at baseline that HOA patients (mean difference: 0.55 [0.17,0.93], p = 0.005), while HOA surgery resulted in a larger decrease in pain intensity than KOA surgery (mean differences at 3 months: 1.462 [1.06,1.87] and 6 months: 1.21 [0.8,1.62], p value <0.001).

The main effect of time on pain intensity showed that from baseline to 3 months there is a large decrease in pain intensity. There was no change in pain intensity between 3 months and 6 months, revealing that pain levels were stable from 3 months onwards in both OA groups (mean differences, KOA: Baseline-3 months 3.8 [3.512,4.124] p<0.001; 3 months-6 months: 0.139 [-1.76,0.45], p = 0.8; HOA: Baseline-3 months 4.73 [4.133,5.332] p<0.001; 3 months-6 months: 0.112 [-0.512,0.736], p = 0.9).

Correlations between pain measure types pre-surgery were significantly positive in both OA groups, generally stronger in KOA than HOA, although these differences were relatively small. At 3- and 6-months post-surgery, the strength of the correlation of pain measures in the HOA group correlations were maintained; however, for the KOA, there was a strengthening of the correlations from baseline. Changes in correlations between pain measures by OA type, post-surgery imply that the characteristics of the pain itself is shifting distinctly post-surgery for each type of OA.

## Pre-surgical pain levels mostly do not relate to post-surgical pain relief

For all 4 pain outcome measures, we examined the relationship between pre-surgical pain and a) residual pain after surgery (100% residual pain meaning no change; 0% residual pain rendering complete relief); b) absolute pain values after surgery, both for KOA and HOA at 3- and 6-months (Fig 3). We observed mostly weak and statistically not significant correlations between pre-surgical pain intensity and residual pain (except BPI pain at 3 months for KOA; NRS at 6 months for HOA; both were weakly negatively but statistically significantly related to pre-surgery values). Similarly, post-surgery pain was weakly and mostly not significantly related to pre-surgery pain (except KOOS at 3 months; and SF36 at 3 and 6 months, for KOA and HOA; all of these were statistically significantly positively related to pre-surgical measures). Both, residual pain and pain, show generally weak relationships with pre-surgery pain, and the relationships are often inconsistent with each other, indicating that pre-surgical pain levels are not consistent predictors of post-surgery measures.

## OA related dimensions

Considering the broad battery of questionnaires and clinical measures collected, we sought to use a data dimensionality reduction approach to define dominant behavioral/clinical factors underlying OA pain and that were subject to change with surgery. To this end, we applied a PCA analysis to the questionnaires and physical performance tests at baseline, focusing on the larger group of KOA patients (n = 84). Pain intensity-related subscales were not included in this analysis, as they are the outcome measures to be modeled by PCA results. The correlations, organized by PCA results, are presented in **Fig 4A**. PCA identified 5 orthogonal components with eigenvalues>1.0, altogether explaining 69.9% of the variance (S2 Fig). Given the observed loadings (S1 Table), we labeled them as: 1) *Affect*, composed of anxiety and

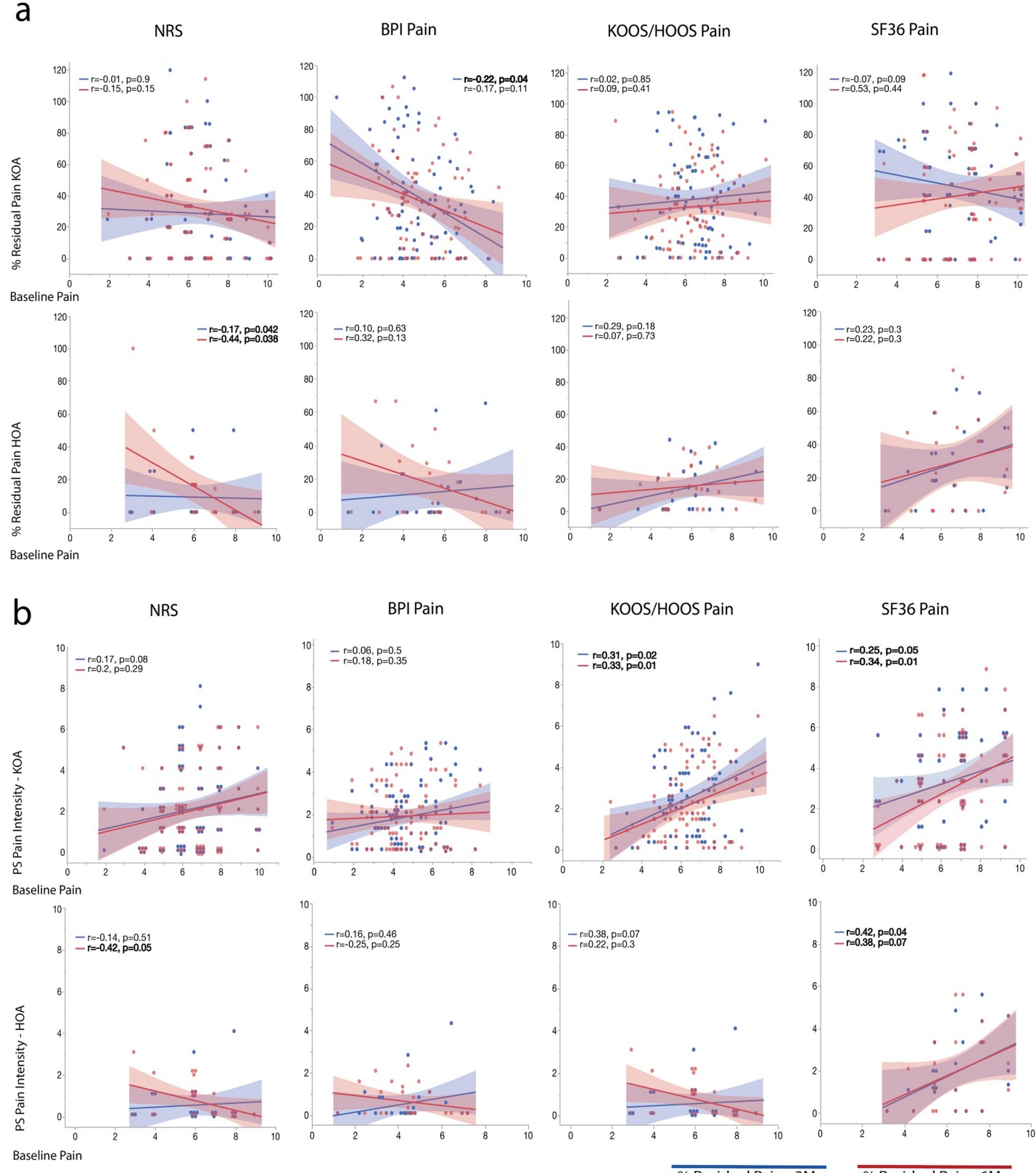

**Fig 3. Influence of baseline pain levels on post-surgical residual pain.** The scatterplots depict patients' percentage residual pain after surgery (% residual pain, where 100% = no change from pre-surgical levels, 0% = full recovery) **(a),** and post-surgery absolute pain intensity **(b)** relative to pre-surgical levels, as a function of pre-

surgical levels, for all four pain outcome measures for KOA and HOA, at 3 (blue) and 6 (red) months post-surgery. Symbols represent subjects. Shaded areas indicate 95% confidence intervals. Results in bold represent statistical significance at p<0.05. BPI Severity, Brief Pain Inventory Pain: severity subscale; HOOS Pain, Hip Injury and Osteoarthritis Outcome Score: pain subscale; NRS, Numeric Rating Scale; SF36 Pain, Short-form (36) Health Survey: pain subscale.

depression subscales of HADS; *2) Pain Catastrophizing*, its highest factor loadings were the three maladaptive dimensions rumination, magnification and helplessness of PCS; 3) *Pain Quality*, dominated by the MPQ-sensory subscale and DN4, with high loadings regarding knee symptoms, knee related quality of life and sports and recreational ability; 4) *Health*, which was dominated by the SF-36 measures that quantify health status and health related quality of life; 5) *Physical Performance*, included high negative loading for 6MWT and high positive loading for TUG (**Fig 4B**). The factors approximate the distinct domains surveyed by the questionnaires and tasks: HADS, KOOS, PCS, SF-36, and the combination of TUG and 6WMT. These five factors were used in subsequent model building to predict pain and residual pain.

## Modelling pain and TJR pain outcomes in OA

Next, we sought to model OA pain, using multi-factorial regressions (including only parameters that survived both forward and backward elimination), both at baseline and after surgery. Independent variables entered in our models are the five factors from the PCA results, together with relevant clinical/demographic variables: age, gender, educational level, body mass index, pain duration, and radiographic severity of OA.

**Pre-surgery KOA pain is defined by its quality, across pain measures.** Pre-surgery KOA pain could be successfully modeled for all four outcome measures (**Table 3**). All models reached statistical significance and accounted for 22–57% of variances of pain intensity. Pain

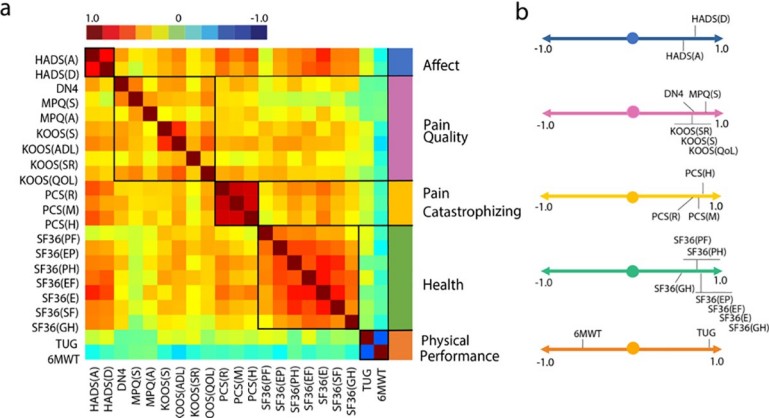

**Fig 4. Principal component analysis identified five factors characterizing baseline KOA.** Pain- and affect-related questionnaires, their subscales, and performance measures (prior to surgery) were examined together to identify dominant underlying factors. **a.** Correlation matrix ordered based on principal component analysis results (Pearson's r represented by color bar). The five identified components were labeled according to membership properties. **b.** Factor loadings are shown for the five components. To highlight dominant factors threshold of factor loading was set on 0.5/-0.5, after Promax oblique rotation. 6MWT, six minute walking test; DN4, The Neuropathic Pain 4 questions; HADS (A), The Hospital Anxiety and Depression Scale, Anxiety; HADS(D), The Hospital Anxiety and Depression Scale, Depression; KOOS, Knee Injury and Osteoarthritis Outcome Score, (ADL–Function in daily living), (S -Knee Symptoms), (SR–Function in sport and recreation), (QOL–knee related quality of life); MPQ, McGill Pain Questionnaire, (A–Affective score) (S–Sensory score); PCS, Pain Catastrophizing Scale, (R–Rumination subscale), (M–Magnification subscale), (H–Helplessness subscale); SF36, Short-form (36) Health Survey, (PF–Physical Functioning), (PH–physical role functioning), (EP–emotional role functioning), (EF–energy/fatigue), (E–emotional well-being), (SF–social functioning), (GH–general health); TUG, (Timed -up and go test).

**Table 3. Multiple regression models for KOA pain intensity at baseline for four different pain intensity measures.**

| Model | b | SE | β | t | p | Adjusted R$^2$ |
|---|---|---|---|---|---|---|
| **BPI Pain Severity** | | | | | | |
| Pain Quality | 2.526 | .670 | .385 | 3.770 | .000 | |
| Pain Catastrophizing | 1.430 | .639 | .229 | 2.239 | .028 | |
| | | | | | | .275**<br>F(2,92) = 18.816 |
| **NRS** | | | | | | |
| Pain Quality | 0.851 | 0.162 | 0.479 | 5.258 | .000 | |
| | | | | | | .229**<br>F(1,93) = 27.652 |
| **KOOS Pain** | | | | | | |
| Pain Quality | 11.499 | 1.126 | .713 | 10.216 | .000 | |
| Physical Performance | 2.426 | 1.123 | .151 | 2.160 | .033 | |
| | | | | | | .573**<br>F(1,92) = 63.379 |
| **SF36 Pain** | | | | | | |
| Health | 10.553 | 1.514 | .602 | 6.970 | .000 | |
| Pain Quality | 3.438 | 1.573 | .189 | 2.186 | .031 | |
| | | | | | | .513**<br>F(2,92) = 50.504 |

KOA baseline pain intensity was explained by different pain-related characteristics, depending on the questionnaire used to capture pain intensity. While the pain quality factor was incorporated in all four regression models, additional unique influences were also identified for three of the four pain intensity scales. Displayed statistics are from the final step of each model. **b**, unstandardized regression coefficient; **SE**, standard error; **β**, standardized regression coefficient; **F**, obtained F-value; **t**, obtained t-value; **R$^2$**, proportion of variance explained.

**p ≤ 0.01. Displayed statistics are from the final step for each dependent variable. BPI Severity, Brief Pain Inventory Pain: severity subscale; HOOS Pain, Hip Injury and Osteoarthritis Outcome Score: pain subscale; NRS, Numeric Rating Scale; SF36 Pain, Short-form (36) Health Survey: pain subscale.

quality emerged as the common dominant factor accounting for higher pain intensity throughout all scales. For NRS it was the only factor present in the model, whereas for the other 3 outcomes, additional factors were identified. BPI severity was predicted by higher levels of Pain Catastrophizing, KOOS Pain by worse Physical Performance, and SF-36 pain by worse Health Status.

**Models predicting pain intensity and residual pain after surgery in KOA.** Next, we sought to model absolute pain intensity after surgery and residual pain (reflecting within subject change from pre-surgery) for all four pain measures, using the parameters collected prior to surgery, thus searching for pre-surgery influences on post-surgical pain. Modeling was restricted to pain at 6 months post-surgery, since there were minimal differences between post-surgery pain at 3 and 6 months.

Results, (Table 4) demonstrated that only three of the four outcome measures for absolute post-surgical pain could be modeled, accounting for 0–24% of the variance, and obtained models were distinct for each pain measure. We obtained similar results when modeling residual pain 6 months post-surgery. Only three of the four pain measures could be modeled, accounting for even smaller 0–11% of the variance, and obtained models were distinct for each outcome measure, as well as from the models obtained for post-surgical pain. Note that obtained results seem paradoxical. Correlations between the four pain outcome measures increases post-surgery yet obtained, pre-surgery based, models diverge from each other, both for pain and for residual pain. A final attempt to model pain intensity as a composite variable

**Table 4. Multiple regression models for post-surgical KOA pain intensity, and for percentage residual pain at 6-months post-surgery, for four different pain intensity measures.**

| Post-surgical Pain Intensity | | | | | | |
|---|---|---|---|---|---|---|
| Model | b | SE | β | t | p | Adjusted R² |
| **BPI Pain Severity** | | | | | | |
| Affect | 2.845 | .670 | .408 | 4.246 | .000 | .238** |
| Pain Duration | .277 | .096 | .278 | 2.893 | .005 | F(2,82) = 13.952 |
| **NRS** | | | | | | |
| No predictive model–no variables entered in the equation. | | | | | | |
| **KOOS Pain** | | | | | | |
| Affect | 5.688 | 2.229 | .284 | 2.522 | .013 | |
| Gender | 9.756 | 4.320 | .221 | 2.259 | .027 | |
| Health | 4.246 | 2.026 | .231 | 2.060 | .043 | |
| | | | | | | .234** F(3,81) = 9.338 |
| **SF36 Pain** | | | | | | |
| Health | 7.032 | 2.292 | .310 | 3.068 | .003 | |
| Gender | 15.176 | 5.520 | .278 | 2.749 | .007 | |
| | | | | | | .196* F(2,82) = 9.730 |
| **% Residual Pain** | | | | | | |
| Model | b | SE | β | t | p | Adjusted R² |
| **BPI Pain Severity** | | | | | | |
| Pain Duration | 1.413 | .536 | .274 | 2.635 | .01 | |
| Health | 7.681 | 3.451 | .232 | 2.226 | 0.029 | |
| | | | | | | .114** F(2,82) = 6.267 |
| **NRS** | | | | | | |
| No predictive model–no variables entered in the equation. | | | | | | |
| **KOOS Pain** | | | | | | |
| Physical Performance | 9.276 | 3.036 | .321 | 3.055 | .003 | |
| | | | | | | .092** F(1,83) = 9.335 |
| **SF36 Pain** | | | | | | |
| Gender | 24.210 | 8.088 | .316 | 2.993 | .004 | |
| | | | | | | .088** F(1,83) = 8.959 |

Different explanatory models were elicited for absolute versus relative (% residual pain) pain intensity after surgery, with the variance explained by each model being overall lower for % residual pain than that for absolute pain intensity. Again, explanatory variables were also distinct considering the four different outcome measures. **b,** unstandardized regression coefficient; **SE,** standard error; **β**, standardized regression coefficient; **F,** obtain F-value; **t,** obtained t-value; **R²**, proportion variance explained. Gender: male coded as 0, female coded as 1.

* p ≤ 0.05

**p ≤ 0.01. Displayed statistics are from the final step for each dependent variable.

BPI Severity, Brief Pain Inventory Pain: severity subscale; HOOS Pain, Hip Injury and Osteoarthritis Outcome Score: pain subscale; NRS, Numeric Rating Scale; SF36 Pain, Short-form (36) Health Survey: pain subscale.

averaging the four outcomes was performed, without informative results and reported in S2 Table.

**Do KOA models of pain and residual pain generalize to HOA?.** Given the smaller data available in HOA (n = 22), and the large number of independent variables and four pain

outcome measures, we limited HOA modeling. We only tested the extent to which variables obtained in KOA modelling are meaningful for HOA. Therefore, regression models were constructed for HOA pre-surgical pain, 6-months absolute post-surgical pain and residual pain using only parameters identified for KOA. Pre-surgery, the multiple regression successfully modeled pain intensity for HOOS Pain (equivalent to KOOS pain), $F_{(2,22)} = 24.308$, $p<0.005$, however only one of the two variables entered, Pain Quality, was significant ($\beta = .764$, $p = 0.005$). For SF-36 pain, the model obtained for KOA was also applicable, $F_{(2,22)} = 23.55$, $p<0.001$. Here the factor Health ($\beta = .732$, $p<0.001$), but not Pain Quality was significant. NRS and BPI in HOA failed to be modeled. For absolute post-surgery pain and residual pain, variables identified on the KOA modelling were not significantly associated with any of the four pain scales in HOA.

### Network analysis of pain dimensions

An alternative to regression-based modeling of the effects of TJR on OA pain is to examine properties of the correlation matrix identified pre-surgery (Fig 4) as a function of type of OA and time from surgery. Representing such correlation matrices as networks provides insights regarding organizational topography and changes in the inter-relationships between pain characteristics that define the OA state, as the variations in individual factor weights can be considered to define the OA-pain personality profile of such patients. Therefore, we calculated these networks pre-surgery, and three- and six-months post-surgery (Fig 5).

Regarding the pre-surgery KOA network, factors Affect, Pain Catastrophizing and Health presented salient edges (significantly high correlations) among them. Pain Quality showed a lower number of edges connecting with other factors (only through subscale KOOS-ADL). Physical performance was segregated from the other factors. For HOA, Affect and Pain Catastrophizing did not share any salient correlations. Pain Quality was highly correlated to Health and to a lesser extent to Pain Catastrophism. Physical performance was again segregated.

At six months after surgery topological differences were identified in both KOA and HOA groups. For the KOA network, Affect and Pain Catastrophizing no longer presented salient edges. Pain quality shared a higher number of edges with Affect and Health. Physical Performance continued to be isolated, sharing no edges with other components. For HOA, Pain Catastrophizing lost its prominent edges with Health and was only linked with Pain Quality. Physical Performance showed links with one variable in Pain Quality (HOOS Sports and Recreational) (Fig 5A).

To quantify topological changes in these network architectures we derived network measures and compared them between groups and as a function of time. We calculated change in strength of connectivity (change in correlation coefficients for all pairs of subscales, Δr-value) both for 3- and 6-months post-surgery. For further comparison intra- and inter-groups, we computed statistical probability using 10,000 permutations with random resampling.

Inside each group, there was a significant change in Δr-value, for both KOA and HOA, at 3 and 6 months, with no differences between 3 and 6 months in each group, indicating that post-surgical connectivity is stable in time. When comparing between KOA and HOA groups, connectivity change was larger for HOA both at 3 and 6 months (Fig 5B).

Lastly, we evaluated the clustering coefficient and modularity of the networks and assessed differences between groups. For both measures, KOA networks remained stable after treatment. HOA, on the other side, showed a significant change in both measures, from baseline to 3 and 6 months. From 3 to 6 months the networks remained stable (Fig 5C).

Overall, we observed that pain characterizing networks for KOA and HOA are quite distinct from each other prior to surgery while displaying similar topology; there is a significant

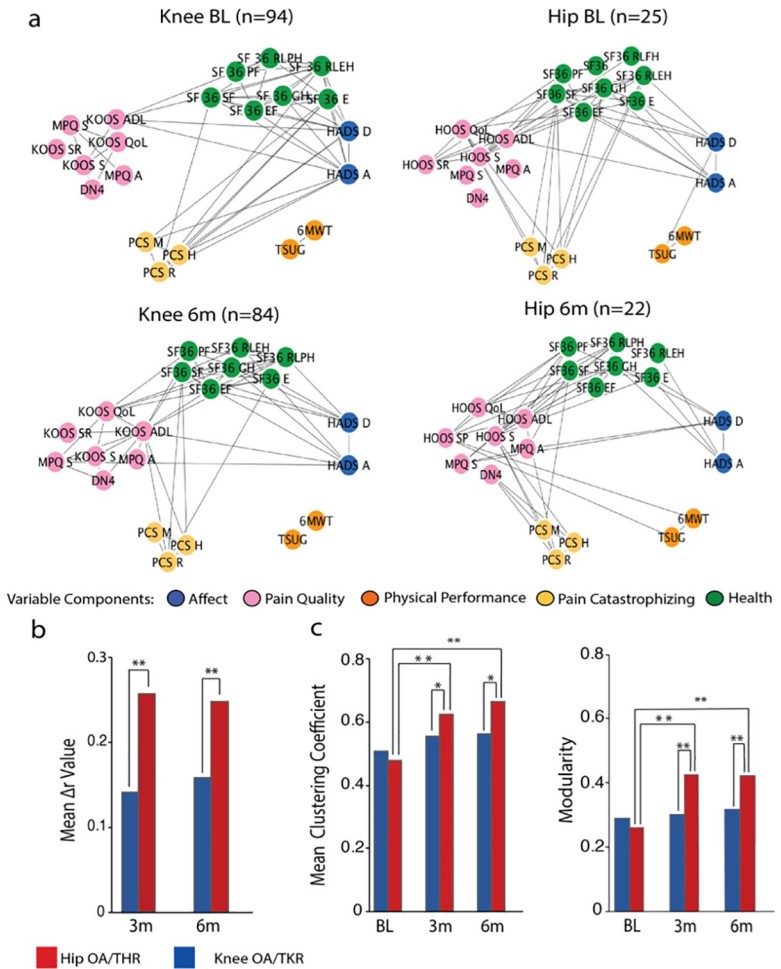

Variable Components: ● Affect ● Pain Quality ● Physical Performance ● Pain Catastrophizing ● Health

**Fig 5. Network representation of OA pain characteristics. a)** Network graphs depict interrelations between clinical and pain-related questionnaire subscale measures at baseline, and at 6 months post-surgery, for KOA and HOA patients. Network communities were derived from the PCA analysis. Links represent the top 25% correlations of each network. **b)** The bar graph displays mean change of global correlation coefficients (Pearson's Δr) for KOA and HOA, at 3- and 6-months post-surgery. Both groups had significant change in the overall interrelations between clinical and pain-related characteristics (KOA mean Δr 3months: 0.14, t = 13.37, mean Δr 6months: 0.16 t = 14.93, HOA mean Δr 3months:0.28, t = 8.72, mean Δr 6months:0.26 t = 9.23, p<0.001). The extent of change remained stable from 3 to 6 months post-surgery and was substantially higher in the HOA group at 3 months (t = 4.62, p<0.001) and 6 months (t = 3.44, p<0.001). **c)** Graph theory-based modularity and mean clustering coefficients for correlation networks at baseline, 3 and 6 months. The HOA networks shows significant topological reorganization 3 months (mcc: t = -8.19, modularity: t = -9.22, p<0.001) and 6 months after surgery (mcc, t = -10.62, modularity, t = -9.02, p<0.001), while KOA remains stable. BL, baseline; 3m, 3 months; 6m, 6 months; Statistical risk probability was computed under 10.000 times repeated random resampling. **p<0.001, *p<0.05.

change in the overall interrelations between clinical and pain-related characteristics after surgery, more profoundly for HOA; and, topological properties show that network reorganization post-surgery is only significant for HOA.

## Discussion

This study examined KOA and HOA pain prior and after TJR surgery. We used a systematic and structured approach, together with data reduction techniques, to investigate the properties of OA pain, its change with surgery, and factors that influence post-surgical OA pain. By using

four distinct pain intensity quantifying measures, two distinct types of joint OA, and measures collected at pre-, 3, and 6 months post-surgery, we examined the contribution of a large number of potential influences, many of which have been reported to be risk factors for OA pain persistence post-TJR. As available data were larger for KOA, we performed model building in this group and tested identified variables in HOA. Each of the four-pain intensity measures we used, demonstrated an overall decrease in OA pain after surgery in both OA groups, that was larger for HOA patients. A striking and perhaps unexpected result was how little OA pain changed from 3- to 6-months post-surgery in both groups. Neither the mean pain nor pain characteristics, as assessed by network properties, showed any important changes over this time period, although large changes were seen between pre-surgery and 3-months post-surgery. Our regression models showed that commonly assessed clinical and behavioral measures prior to surgery fail to reliably predict pain outcomes after TJR.

OA pain and persistent pain after TJR have been previously studied using multiple pain outcome scales. These can be divided in two major groups, general measures such as NRS, visual analog scale, SF-36 bodily pain and BPI pain severity, and OA specific measures as Western Ontario and McMaster Universities Osteoarthritis Index (WOMAC), KOOS/HOOS pain score and the Oxford Knee Score pain subscale [7, 35]. Such studies suggest that different pain outcomes relate to different facets of the pain experience in knee OA [36]. Using four different pain intensity outcomes, three of them from the category of general pain scales and one specific for OA (HOOS/KOOS), our results show that although correlations between these measures are positive and mostly significant (both at baseline and post-surgery), BPI pain severity tends to underestimate pain intensity and SF-36 pain tends to overrate pain intensity after surgery, both in KOA and HOA groups. Still, all four measures decreased 3-months post-surgery, and all remained unchanged between 3- and 6-months post-surgery.

Pain outcomes concerning persistency are commonly studied using primarily the absolute value of pain intensity after surgery, or dichotomizing the outcome using a fixed threshold that varies across studies [37–40]. Such approaches assume that the treatment has a constant effect. A change may show the health improvement in a more observable way. Here we chose to use both change (residual pain) and absolute value of pain. Both measures showed independence or minimal and inconsistent dependence on baseline values, implying pain relief post-surgery does not depend or inconsistently depend on entry scores.

An important remark concerning post-surgical pain and its risk prediction is that it relies on how it is defined and thus also on how one measures the pain outcomes. Chronic post-surgical pain is accepted as the pain that persists at least three months after surgery, different in characteristics from pre-operative pain, and without other causes such as infection or technical failure [5]. Our results are generally consistent with this definition and further advance the concept. Firstly, we observe that models characterizing OA pain at baseline do not generalize to pain post-surgery. Second, the amount of variance explained with the regression models for pain intensity decreased from pre-surgery (accounting for 23–57% of pain intensity variance), to post-surgery (accounting for 20–24% of variance), and further decreased when modeling residual pain (accounting for 9–11% of variance). Given that residual pain is a more direct measure of the influence of the surgical intervention than the absolute value of pain intensity, our models at best could only explain 11% of the variance of the surgery related OA pain. Third, studies report that pre-operative OA pain intensity has a strong influence on post-surgical outcomes [7]. It was recently argued that the evidence for this influence is of low-quality, even when studied in much larger number of OA patients [12], and our results support the failure of pre-operative pain as a predictor of post-surgical outcomes. Fourth, the network analysis shows large changes in the interrelationships between pain related characteristics post-surgery. Thus, our analysis, especially for KOA where we examined multiple models,

suggests that the post-operative pain is minimally related to the pre-operative pain properties. Our results in HOA, although not as strong, are also consistent with this notion.

Given the small sample size in HOA, we limited the statistical tests in this group. Models derived from KOA did not yield significance in the HOA group. Thus, HOA pain models remain to be studied in larger data sets in the future, and with additional parameters not included here. However, our results repeatedly confirm that pain relief is better in this group and this is accompanied with larger changes in the network properties. We observed larger changes in clustering coefficient and in modularity in HOA, implying that the pain personality in HOA is being fractured with pain relief, rendering different factors independent from each other. These findings are all consistent with earlier reports showing that the improvement in pain and physical function after arthroplasty is greater for hip than knee OA [3], even though symptomatic presentation of HOA is associated with more advanced radiological disease [41]. Determinants for persistent pain after THR are less studied, and evidence is limited and conflicting [42]. The full scope of the differences in TJR outcomes between both conditions requires further studies.

The primary focus of this paper was to find predictors for pain and pain persistency, and although we show that there is a high variability concerning scales and outcome definitions, some of the findings deserve further discussion. At baseline, we observed that across the four scales and the aggregated pain measure, Pain Quality (constituted mainly by neuropathic pain profile and sensory quality of MPQ) related to higher pain intensity. When we modelled residual pain after surgery, each scale unveiled different predictors. No homogeneous result could be retrieved.

It has been reported that the greatest improvement in patients undergoing TJR happens in the first 3 months after surgery [3]. Although a precise timeline for pain recovery is difficult to draw, our results support the finding that pain persistence at 3 months should be regarded as critical evidence for longer-term persistence of post-surgical pain.

An important weakness of the present study is the relatively small sample size and large number of variables tested. We acknowledge the increasing likelihood of type 2 errors and consider it as an important limitation that should be highlighted. It is possible that in larger samples stronger statistical relationships may be uncovered between presurgical clinical measures and post-TJR pain. However, we should note the literature where larger groups of subjects were studied indicate that these relationships are small in magnitude, and thus of debatable biological interest [12]. Nonetheless, we believe our results should be interpreted cautiously regarding generalizability, and the use of current methodology in larger samples would be of interest in the future.

The topic of reliability and sample size was recently discussed [43] in the field of neuroimaging but rendering important implications to other areas. The authors point that while sample size is recognized as a major determinant of statistical power, measurements of reliability are less commonly considered, that place an upper limit on the maximum detectable effect size. We measured inter-reliability regarding the radiographic assessment and used questionnaires with high context validity, internal consistency and test-retest reliability. Although we did not measure reliability directly, we believe this is an important remark, and data quality assessments are always important.

Another limitation is the imbalance of available data between KOA and HOA. Moreover, there were important demographic differences between the two groups which could not be corrected for due to the limited available sample in HOA. Thus, we cannot rule out the influence of these factors on the models derived from KOA and tested in HOA.

Regarding our sample characteristics, patients were enrolled in the same center, and the population included is ethnically homogeneous, thus caution is needed in generalizing the

present's study results to other populations. The follow-up time was limited to 6 months, what can also be regarded as a limitation. We also did not collect multiple measures that in the literature have been suggested to influence both baseline pain and post-surgical pain. For instance, measures of widespread hypersensitivity, temporal summation of pain and impaired endogenous pain inhibition assessed by quantitative sensory testing, have been suggested to contribute to poor pain relief following TKR [44, 45], however see [46]. It was also previously shown that OA patients present central nervous system structural and functional maladaptive changes [47, 48]. We will test the latter concept in this same group of participants using their brain imaging results.

In conclusion, our results show distinct pain scales relate to different aspects of the pain experience. Post-surgery residual pain scores show primarily independence from baseline pain. There is a reorganization of pain related biopsychosocial parameters that define the OA personality, and this change seems more profound in hip OA where pain relief is also larger.

## Supporting information

**S1 Fig. Pain intensity at baseline, 3 and 6 months for KOA and HOA.** Interaction between the 2 OA groups, time (baseline, 3 months and 6 months), and measurement type (information present in Table 3). HOA patients presented larger pain relief than KOA. For both groups, pain intensity ratings did not show meaningful differences between 3- and 6-months post-surgery. Mo, months.
(TIF)

**S2 Fig. Principal component analysis on correlations of pain related measures. a.** Scree plot and percentage of variance explained by each component. The eigenvalues become lower than 1.0 at the fifth component, and the slope of eigenvalues flattens at this component. Components were retained only for eigenvalues higher than 1, corresponding to a percentage of variance higher than 5%. A total variance of 69.54% was explained by the 5 selected components
(TIF)

**S1 Table. Principal Component Analysis–Factor loadings.** Threshold of factor loading was set on 0.5/-0.5 after Promax oblique rotation (bold). 6MWT = 6 minute walking test; DN4 = The Neuropathic Pain 4 questions; HADS(A) = The Hospital Anxiety and Depression Scale, Anxiety; HADS(D) = The Hospital Anxiety and Depression Scale, Depression; KOOS = Knee Injury and Osteoarthritis Outcome Score, (ADL–Function in daily living), (S -Knee Symptoms), (SR—Function in sport and recreation), (QOL—knee related quality of life); MPQ = McGill Pain Questionnaire, (A–Affective score) (S–Sensory score); PCS = Pain Catastrophizing Scale, (R–Rumination subscale), (M–Magnification subscale), (H–Helplessness subscale); SF36 = Short-form (36) Health Survey, (PF–Physical Functioning), (PH–physical role functioning), (EP–emotional role functioning), (EF–energy/fatigue), (E–emotional well-being), (SF–social functioning), (GH–general health); TUG = Test stand-up and go.
(DOCX)

**S2 Table. Multiple regression analysis for KOA pain intensity and % residual pain at 6-months post-surgery, using an aggregated variable for pain intensity (average of four pain intensity questionnaire measures).** A composite measure of pain intensity was built averaging the four outcome scales. Prediction models of our aggregate pain intensity measure show differences across absolute and relative measures (% residual pain). Physical performance was a common predictive factor of both measures. No large effect size was captured in these models, rendering the 4 measures together are not capturing important predictive information. **b,** unstandardized regression coefficient; **SE** standard error; **β**, standardized regression

coefficient; **F**, obtain F-value; **t**, obtained t-value; $R^2$, proportion variance explained. All statistics are from the final step of the model. $^{**}p \leq 0.01$. As the pre-surgical parameters predicting post-surgical pain or residual pain for four pain outcome measures captured distinct independent variables, we reasoned that each may be reflecting specific characteristics and thus combining all four measures would predict larger variance and incorporate the component characteristics. Therefore, we constructed the composite, average score, of all four pain outcome measures and studied its properties. We tested how pre-surgical factors predict 6-months post-surgical KOA pain, using our aggregate measure (S2 Table), again modeling pain and residual pain for KOA patients. For post-surgical aggregated pain severity, the model explained 19% of the variance and included worse health state, lower degree of structural articular damage, and poor results in the physical performance tests. For residual pain, the model explained 7% of the variance and Physical Performance was the only predictive factor. Using these variables to predict HOA post-surgical pain and residual pain we could not find any statistically significant models.

(DOCX)

## Acknowledgments

We thank to Dr. Paulo Oliveira, for the clinical support and supervision of patients enrolled in the study and all the members of the Orthopedic Department of Centro Hospitalar Universitário de São João, Porto, for their support with data collection. Dr. Lina Melão and Dr. Patricia Leitão for the assistance in the radiographic classification. We are thankful to all Apkarian lab and Galhardo lab members that contributed to this study with their time and resources.

## Author Contributions

**Conceptualization:** Joana Barroso, Vasco Galhardo, A. Vania Apkarian.

**Data curation:** Joana Barroso, João Pinto-Ramos.

**Formal analysis:** Joana Barroso, Kenta Wakaizumi.

**Investigation:** Joana Barroso.

**Methodology:** Joana Barroso, Vasco Galhardo, A. Vania Apkarian.

**Resources:** João Pinto-Ramos.

**Supervision:** A. Vania Apkarian.

**Writing – original draft:** Joana Barroso.

**Writing – review & editing:** Joana Barroso, Kenta Wakaizumi, Diane Reckziegel, Thomas Schnitzer, Vasco Galhardo, A. Vania Apkarian.

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
