## [Decision Letter · Decision Letter 0]

17 Oct 2019

PONE-D-19-23720

Prognostics for pain in osteoarthritis: Do clinical measures predict pain after total joint replacement?

PLOS ONE

Dear professor Apkarian,

Thank you for submitting your manuscript to PLOS ONE. After careful consideration, we feel that it has merit but does not fully meet PLOS ONE’s publication criteria as it currently stands. Therefore, we invite you to submit a revised version of the manuscript that addresses the points raised during the review process.

We would appreciate receiving your revised manuscript by Dec 01 2019 11:59PM. To enhance the reproducibility of your results, we recommend that if applicable you deposit your laboratory protocols in protocols.io, where a protocol can be assigned its own identifier (DOI) such that it can be cited independently in the future. For instructions see: http://journals.plos.org/plosone/s/submission-guidelines#loc-laboratory-protocols

We look forward to receiving your revised manuscript.

Kind regards,

Yuanyuan Wang

Academic Editor

PLOS ONE

Additional Editor Comments:

While this study would answer an important research question, the reviewers have raised concerns about the statistical analysis.

There are some further comments from the reviewer 3.

1. The relationship among sample size, measurement reliability, statistical power and effect size has been recently demonstrated in a publication (see https://www.nature.com/articles/s41562-019-0655-x). It is thus critical to report the reliability of all the measurements employed in the manuscript. It would be perfect if the authors can test the reliability directly, otherwise the citation on the previous reliability publication is fine.

2. So many statistical tests are performed, which should be corrected in some forms for multiple comparisons. For examples, FDR methods.

3. It is a problem that many tests were done with such a small sample size, which limited the generalizability of the current findings. This must be discussed as a limitation.

Reviewers' comments:

Reviewer's Responses to Questions

**Comments to the Author**

1. Is the manuscript technically sound, and do the data support the conclusions?

Reviewer #1: Partly

Reviewer #2: Yes

Reviewer #3: Partly

2. Has the statistical analysis been performed appropriately and rigorously? 

Reviewer #1: I Don't Know

Reviewer #2: I Don't Know

Reviewer #3: No

3. Have the authors made all data underlying the findings in their manuscript fully available?

Reviewer #1: No

Reviewer #2: Yes

Reviewer #3: No

4. Is the manuscript presented in an intelligible fashion and written in standard English?

Reviewer #1: Yes

Reviewer #2: Yes

Reviewer #3: Yes

5. Review Comments to the Author

Reviewer #1: The authors aim to identify pre-operative variables associated with residual pain after replacement surgeries for knee and hip. The key limitation is the small sample size for a large number of variables. There were numerous statistical methodologies utilized to handle the above problem. They have shown nicely that post-surgery residual pain was not associated with baseline pain. They found different types of pain were associated with post op pain, but failed to identify variables associated with post-op pain in multivariable analysis. It is unclear whether it is the absence of predictive variables, or just related to sample size and high variability in the pre-op variables.

• Please clarify how many patients (each for KOA / HOA) that have complete set of data, how many were imputed, and how many were excluded due to missing data.

• It is not clear what were illustrated in Table 1. To me, it should be the 84 KOA and 22 HOA subjects included in the study. Please specify the number of patients clearly in table 1.

• Labels of figure 3 are not clear. There are 4 figures for (a) and 4 figures for (b). I am guessing that they represent the 4 types of pain. These need to label properly. It is also not sure what was indicated in the X-axis and Y-axis of all the figures.

• For table 2, authors did correlation between 4 types of pain before and after surgery. This part was not described in the methods. Please describe.

• Authors conducted a PCA to identify variables in clusters. The results of the PCA just shown in figure 4 is not easy to understand and need tremendous amount of explanation. The loadings of each variable onto the 5 factors must be illustrated. How were the 5 components concluded from the PCA? What really constitute the 5 components: affect, pain quality, pain catastrophizing, health, physical performance. It is unclear why things measuring similar concepts were not clustered together, such as SF-physical functioning, physical performance, KOOS function. Other things expected to cluster are SF-mental health, HADS, pain catastrophizing. But these were dispersed in the 5 components instead, please explain.

• In the multi-variable regression analysis, apart from the 5 components identified in the PCA, the authors also considered age, gender, educational level, body mass index,

• pain duration, and radiographic severity of OA. I wonder why these other factors were not included in the PCA? Then what may be the true purpose of the PCA?

• I would advise the authors to skip the regression models in HOA, given the reasons stated by the authors: small sample size (n=22), large number of independent variables, and four pain outcome measures. Given the small sample size, even the 5 components identified from the PCA would be too many. Limiting the regression model to components or variables from the KOA models would not prove or disprove whether HOA models conform to the KOA models. Therefore, these analysis are futile and should be aborted.

• The analysis of the composite is debatable. The “composite” was created out of the blue from averaging the 4 types of pain. The BPI is more of neuropathic, which SF-BP, NRS and KOOS pain were more about pain severity and pain related to activity. Therefore the composite will theoretically driven by 3 scores that were severity and functional related. As the regression model confirm. But it really depend on what you put into the formula as “composite”. Therefore, this type of “composite” without any conceptual framework is rather meaningless.

• Discussion. The key limitation is the tiny sample size and the consideration of numerous pre-operative variables. This has to be highlighted.

Reviewer #2: This is an interesting study describing both predictors and the course of pain prior to and post Total Joint Replacement in Knee en Hip osteoarthritis. The total sample size (n=120) is not very large for an observational study. The study authors follow an exploratory data analysis strategy while applying many data analyses techniques ranging from linear regression analyses and ANOVA to principal components analyses and network analyses. The results indicate that clear prognostic indicators for pain persistence are lacking. The results are in line with other studies.

This paper is not easy to read in particular for those who are not familiar with all these statistical analyses techniques (for example busy surgeons or other clinicians). The study objectives are broad and lack focus (‘we test the hypothesis….., we examine how …, we attempt to develop …, we assess the reorganization of ….’). It is no surprise that with this list of objectives the study authors end up with a paper presenting many analyses techniques and lots of data in an incoherent way (I am sorry to say that, I have no concerns about the way the study authors applied the analyses techniques but the average reader is probably overwhelmed with too much information). My advice would be to simplify the analyses part or to spread the analyses and results over multiple papers.

Some further comments:

- For data imputation multiple imputation is currently the preferred imputation method.

- Descriptive statistics of the course of the different variables over time (at the four time points) are lacking while this may provide valuable insight to the reader.

- The STROBE guideline for reporting observation studies may be helpful to improve the clarity of this paper.

- The ability to replicate study findings is very important. I am wondering why applying a PCA to the total questionnaire (resulting in new constructs/variables) was preferred over using the existing scales since they have been validated previously.

Reviewer #3: The authors evaluated knee and hip OA patients before, 3 and 6 months post-TJR searching for clinical

predictors of pain persistence. It is nice to see that the reliability is reported for radiographic assessment. However, reliability is extremely important for all the clinical applications of all the assessments. It would be a value of including an independent paragraph as part of the Introduction to highlight the topic on reliability (see and cite this recent commentary https://www.nature.com/articles/s41562-019-0655-x). Especially, the whole project included brain imaging with multiple modalities such as T1, rs-fMRI and DTI although not reported in the present work. On the other hand, the reliability of other assessments reported in the present manuscript should also be presented in the manuscript.

6. PLOS authors have the option to publish the peer review history of their article (what does this mean?). If published, this will include your full peer review and any attached files.

Reviewer #1: No

Reviewer #2: Yes: Bart Staal

Reviewer #3: No

---

## [Author Response · Author response to Decision Letter 0]

19 Nov 2019

Response to Reviewers

Reviewer 1

Reviewer #1: The authors aim to identify pre-operative variables associated with residual pain after replacement surgeries for knee and hip. The key limitation is the small sample size for a large number of variables. There were numerous statistical methodologies utilized to handle the above problem. They have shown nicely that post-surgery residual pain was not associated with baseline pain. They found different types of pain were associated with post op pain but failed to identify variables associated with post-op pain in multivariable analysis. It is unclear whether it is the absence of predictive variables, or just related to sample size and high variability in the preop variables.

Response: We appreciate the Reviewer positive evaluation and take into consideration the concerns regarding the absence of predictive variables across pain scales. Given the lack of robust predictors of pain persistence after TJR in the literature, we undertook a large effort to better uncover this issue. The sample size is indeed smaller than many reports in literature regarding this topic. This is a limitation that we cannot overcome at this point. However, it would be difficult to determine the sample size in advance without a hypothesis concerning the link between each one of the main predictive factors and post-surgical pain scales. Moreover, this is an exploratory analysis, and we are looking for large effect sizes. We believe that if strong predictors cannot be found across distinct scales in a sample with this size, it is improbable that a larger sample size would be more informative. Additionally, we model outcomes as continuous variables, instead of binarizing or categorizing them, which adds precision to our analysis. 

We believe that the evaluation of multiple scales and the novelty of methodology here applied shed light into important and urgent questions to the topic and should in fact be tested in distinct and larger samples in the future. 

We added a paragraph about reliability, sample size and statistical power in the discussion, as suggested by reviewer #3, question 3.1; we further comment the limitations of the sample size in the discussion.

Please clarify how many patients (each for KOA / HOA) that have complete set of data, how many were imputed, and how many were excluded due to missing data.

Response: We appreciate this comment and added more detail on how missing data was handled. Data on patients who withdrew from the study were not used. Of all 84 KOA and 22 HOA patients included, a complete dataset was available for all patients and missing data was scarce – 8 of 102 patients had missing data on one or more questionnaires, and a maximum of 12% of questions was missing for each inputted questionnaire. This information was added (lines 348-351): 

“Most included patients had a complete dataset, except 8% (n=8) had missing data for at least one questionnaire and for those no more than 12% of each questionnaire was missing. The missing data were imputed using the mean for each scale/subscale.”

It is not clear what were illustrated in Table 1. To me, it should be the 84 KOA and 22 HOA subjects included in the study. Please specify the number of patients clearly in table 1.

Response: The number of patients (84 KOA; 22 HOA) were included in the table (first row) and repeated in the table caption. We also clarified that this was the total number of patients included in the analysis (line 306). 

Labels of figure 3 are not clear. There are 4 figures for (a) and 4 figures for (b). I am guessing that they represent the 4 types of pain. These need to label properly. It is also not sure what was indicated in the X-axis and Y-axis of all the figures.

Response: This figure was revised; labels were placed for each plot and information on axis indicated. 

Figure 3. Influence of baseline pain levels on post-surgical residual pain. 

The scatterplots depict patients’ percentage residual pain after surgery (% residual pain, where 100% = no change from pre-surgical levels, 0% = full recovery) (a), and post-surgery absolute pain intensity (b) relative to pre-surgical levels, as a function of pre-surgical levels, for all four pain outcome measures for KOA and HOA, at 3 (blue) and 6 (red) months post-surgery. Symbols represent subjects. Shaded areas indicate 95% confidence intervals. Results in bold represent statistical significance at p<0.05. BPI Severity, Brief Pain Inventory Pain: severity subscale; HOOS Pain, Hip Injury and Osteoarthritis Outcome Score: pain subscale; NRS, Numeric Rating Scale; SF36 Pain, Short-form (36) Health Survey: pain subscale. 

For table 2, authors did correlation between 4 types of pain before and after surgery. This part was not described in the methods. Please describe.

Response: This is now further described in the methods: “Interrelationship of the primary outcome variables (all scored on a 0-10 score) was assessed through correlation analysis using Pearson product-moment tests. Fischer’s z tests were used to evaluate differences between correlation coefficients at baseline, 3 and 6 months.” (line 267).

Authors conducted a PCA to identify variables in clusters. The results of the PCA just shown in figure 4 is not easy to understand and need tremendous amount of explanation. The loadings of each variable onto the 5 factors must be illustrated. How were the 5 components concluded from the PCA? What really constitute the 5 components: affect, pain quality, pain catastrophizing, health, physical performance. It is unclear why things measuring similar concepts were not clustered together, such as SF-physical functioning, physical performance, KOOS function. Other things expected to cluster are SF-mental health, HADS, pain catastrophizing. But these were dispersed in the 5 components instead, please explain.

Response: The PCA loadings are illustrated in figure 4.b – (caption figure 4, line 447: b. Factor loadings are shown for the five components.) This figure is showing the factors with loadings higher than 0.5 or lower than - 0.5. This is described in the methods: line 256: “Threshold for component retention was set on eigenvalues >1.0, together with visual inspection of the scree plot for evaluation of the inflection point. A factor rotation on the obtained components was applied using a Promax oblique rotation technique. Threshold of factor loading was set on 0.5/-0.5 and components were labeled given the observed loadings.” 

However, we agree that this is not totally clear in the results. 

We changed the caption of figure 4.b : “b. Factor loadings are shown for the five components. Threshold of factor loading was set on 0.5/-0.5 after Promax oblique rotation.”

Additionally, we add a supplementary figure illustrating the PCA analysis results, eigenvalues for each component, and variance explained (Supplementary Figure 1), and a supplementary table illustrating the loading for each component (Supplementary Table 1). 

Supplementary Figure 1. Principal component analysis on correlations of pain related measures. a. Scree plot and percentage of variance explained by each component. The eigenvalues become lower than 1.0 at the fifth component, and the slope of eigenvalues flattens at this component. Components were retained only for eigenvalues higher than 1, corresponding to a percentage of variance higher than 5%. A total variance of 69.54% was explained by the 5 selected components.

Supplementary Table 2. Principal Component Analysis – Factor loadings.

 Factor 1 Factor 2 Factor 3 Factor 4 Factor 5

 Health Pain

Quality Pain Catastrophism Physical Performance Affect

HADS(A) 0.1716056 0.1662774 0.0878845 0.0725140 0.7154581

HADS(D) 0.2974458 0.0646839 0.0542344 0.1274329 0.5458700

DN4 -0.0216157 0.6065968 0.0700990 -0.1300625 0.1298646

MPQ(S) -0.2259922 0.7435562 0.0818952 -0.1135575 0.0112932

MPQ(A) 0.0929373 0.3093876 0.1553277 -0.1234311 0.0713686

KOOS(S) 0.1247400 0.6513952 -0.0126017 0.1103185 0.0985363

KOOS(ADL) 0.1706909 0.3895259 -0.0740881 0.1685821 0.0088291

KOOS(SR) 0.0884324 0.6775176 0.0965312 -0.0413376 -0.0768334

KOOS(QOL) 0.1352021 0.6553364 -0.1002832 0.1361084 0.0490056

PCS(R) 0.1151239 0.0055719 0.8425709 0.0324509 0.0269198

PCS(M) 0.0182493 0.0609084 0.9334730 0.0334168 -0.0905804

PCS(H) -0.0604199 0.0307351 0.8566090 0.0265412 0.1380779

SF36(PF) 0.6733634 0.1644234 0.0196261 0.1004566 -0.4028870

SF36(PH) 0.7094073 0.1186078 -0.0392796 0.0892508 -0.0624895

SF36(EP) 0.7884761 0.0366737 0.0977453 -0.1750177 -0.0207307

SF36(EF) 0.7753548 -0.0168216 -0.0386067 0.1019787 0.1113146

SF36(E) 0.7992535 -0.1564045 0.1015777 -0.1445349 0.2852896

SF36(SF) 0.7536360 0.0028377 0.0181402 -0.0697790 0.0637001

SF36(GH) 0.5381934 -0.0661540 -0.0300044 -0.0018053 0.2523995

TUG -0.0508260 -0.1175723 0.1034888 0.9309554 -0.0030743

6MWT 0.0403210 -0.0500505 0.0346415 -0.6220702 -0.1355920

Supp Table 2. Principal Component Analysis – Factor loadings. Threshold of factor loading was set on 0.5/-0.5 after Promax oblique rotation (bold). 6MWT= 6 minute walking test; DN4= The Neuropathic Pain 4 questions; HADS(A) = The Hospital Anxiety and Depression Scale, Anxiety; HADS(D)= The Hospital Anxiety and Depression Scale, Depression; KOOS = Knee Injury and Osteoarthritis Outcome Score, (ADL – Function in daily living), (S -Knee Symptoms), (SR - Function in sport and recreation), (QOL - knee related quality of life); MPQ = McGill Pain Questionnaire, (A – Affective score) (S – Sensory score); PCS= Pain Catastrophizing Scale, (R – Rumination subscale), (M – Magnification subscale), (H – Helplessness subscale); SF36 = Short-form (36) Health Survey, (PF – Physical Functioning), (PH – physical role functioning), (EP – emotional role functioning), (EF – energy/fatigue), (E – emotional well-being), (SF – social functioning), (GH – general health); TUG = Test stand-up and go.

We believe the reviewer’s point about variable representation in each factor is important. From a theoretical perspective, we could expect that features relating to the same sphere (Eg. SF36- Emotional well-being and HADS; KOOS function and SF-36 physical health) condense together given the probable high correlation. In fact, the correlation matrix shows these relationships (Figure 4.a). However, the PCA groups these parameters somewhat differently (an unsupervised learning methodology), with the number of components chosen using robust accepted criteria (Kaiser criteria, eigenvalue and explained variance), and the oblique rotation was performed to define the coefficients of linear regression. No other exploration of the data was performed after obtaining the factor loadings. Given the displayed correlation matrix, we can only predict that if a different number of components were to be chosen, a slightly distinct factor loading in each component would be achieved. Three of the components are mainly constructed upon measures that are subscales of questionnaires as stated in line 447: “Note the factors approximate the distinct domains surveyed by the questionnaires and tasks”. We believe our result is thus interpretable, enabling retaining trends and patterns of the data while reducing dimensionality. 

In the multi-variable regression analysis, apart from the 5 components identified in the PCA, the authors also considered age, gender, educational level, body mass index,

pain duration, and radiographic severity of OA. I wonder why these other factors were not included in the PCA? Then what may be the true purpose of the PCA?

Response: We believe this is an important issue and thus should be clarified. The PCA was applied solely to clinical factors evaluated with questionnaires and physical performance tasks. This were the measures further evaluated in the network analysis and the ones subject to change with the intervention. The goal of the PCA was to simplify the high dimensionality of these measures, while retaining its patterns. Demographic measures, duration of pain and radiographic severity are previously indicated as possible predictors of pain outcomes after TKR and thus introduced in the regression analysis, together with the components obtained from the questionnaire and physical performance tasks after data reduction. 

The first sentence of the results was changed to better characterize this rationale: “Considering the broad battery of questionnaires and clinical measures collected, we sought to use a data dimensionality reduction approach to define dominant behavioral/clinical factors underlying OA pain, and that were subject to change with surgery.”

I would advise the authors to skip the regression models in HOA, given the reasons stated by the authors: small sample size (n=22), large number of independent variables, and four pain outcome measures. Given the small sample size, even the 5 components identified from the PCA would be too many. Limiting the regression model to components or variables from the KOA models would not prove or disprove whether HOA models conform to the KOA models. Therefore, these analyses are futile and should be aborted.

Response: We partially agree with the reviewer statement and believe we were not clear in the description of the methodology applied. Here, given the limitations stated above, we did not enter the 5 components obtained previously for modelling each pain outcome in HOA. We selected, for each scale, meaningful variables captured with the KOA modelling (between 1-3 variables for each model) and tested if there was an association between those for HOA group. We are not attempting to compare the two conditions but simply explore to which extent HOA group follows a similar pattern across pain measurements. We acknowledge and state the limitations of this analysis, however, believe its purpose is meaningful and may add value to an important but often underdiscussed topic. 

We clarified the methodology applied, better characterized the obtained results and reviewed the wording used regarding this topic. We believe we are cautious while arguing the findings in the discussion. 

The analysis of the composite is debatable. The “composite” was created out of the blue from averaging the 4 types of pain. The BPI is more of neuropathic, which SF-BP, NRS and KOOS pain were more about pain severity and pain related to activity. Therefore the composite will theoretically driven by 3 scores that were severity and functional related. As the regression model confirm. But it really depends on what you put into the formula as “composite”. Therefore, this type of “composite” without any conceptual framework is rather meaningless.

Response: We appreciate the Reviewer opinion in this topic. We agree with this view, the analysis of the composite variable was thus excluded from the manuscript and the regression analysis of the composite outcome added as supplementary material. 

Discussion. The key limitation is the tiny sample size and the consideration of numerous pre-operative variables. This has to be highlighted.

Response: We appreciate this suggestion. A more throughout discussion relating to this point was added to the discussion. 

Reviewer 2

Reviewer #2: This is an interesting study describing both predictors and the course of pain prior to and post Total Joint Replacement in Knee en Hip osteoarthritis. The total sample size (n=120) is not very large for an observational study. The study authors follow an exploratory data analysis strategy while applying many data analyses techniques ranging from linear regression analyses and ANOVA to principal components analyses and network analyses. The results indicate that clear prognostic indicators for pain persistence are lacking. The results are in line with other studies.

This paper is not easy to read in particular for those who are not familiar with all these statistical analyses techniques (for example busy surgeons or other clinicians). The study objectives are broad and lack focus (‘we test the hypothesis….., we examine how …, we attempt to develop …, we assess the reorganization of ….’). It is no surprise that with this list of objectives the study authors end up with a paper presenting many analyses techniques and lots of data in an incoherent way (I am sorry to say that, I have no concerns about the way the study authors applied the analyses techniques but the average reader is probably overwhelmed with too much information). My advice would be to simplify the analyses part or to spread the analyses and results over multiple papers.

Response: We welcome the Reviewer positive assessment of the manuscript and take into consideration the issue of the density and complexity of the analysis for the average reader. Considering the lack of robust predictors for post-TJR pain in the literature over the last decades, and the importance of the topic, we chose to conduct a systematic and throughout analysis, studying distinct pain metrics and applying distinct techniques. In order to report the results transparently we chose to be transparent in the data presentation. The concern of Reviewer regarding the complexity of the analysis and density of the paper is well taken in this reviewed version:

- We reviewed the introduction and hypothesis stated, trying to make clear the main goals of the paper. 

- We included statements summarizing the reported observations when lacking.

- Improved the clarity of the language.

- Excluded part of the analysis: composite variable (section 5.3.4), as pointed by the Reviewer #1. 

Some further comments:

- For data imputation multiple imputation is currently the preferred imputation method.

Response: We agree with the Reviewer. The imputation technique used was not highlighted during the analysis as missing data was scarce. Only 8 of 102 patients had missing data on one or more questionnaires, and a maximum of 12% of questions was missing for inputted questionnaires. As requested by #Reviewer1 this information was added.

We believe, given the paucity of missing data, analysis results would not be different regarding the imputation method used. However, hereby we tested the correlation for all studied variables between the two methods (using multiple imputation and mean imputation to the 8 subjects with missing data). All correlations ranged between 0.96 and 1.0, p<0.001. 

- Descriptive statistics of the course of the different variables over time (at the four time points) are lacking while this may provide valuable insight to the reader.

Response: We added a figure in Supplementary Material illustrating the pain intensity per scale, time and type of OA, corresponding to the information available in Table 2. 

- The STROBE guideline for reporting observation studies may be helpful to improve the clarity of this paper.

Response: We assessed the STROBE guidelines and adjusted the paper in regard. Together with this report we add the guidelines with location (page), for each STROBE requirement. 

- The ability to replicate study findings is very important. I am wondering why applying a PCA to the total questionnaire (resulting in new constructs/variables) was preferred over using the existing scales since they have been validated previously.

Response: This is an important point. PCA was applied with two main purposes: 1) Perform data reduction previously to regression analysis, in order to reduce the number of predictor variables; 2) provide a better conceptual understanding of the underlying data. PCA allows us to reduce the dimensions of the dataset, increase interpretability but at the same time minimizes information loss. 

We better explain the rationale for using PCA in the methods: “Due to the high number of clinical and psychological measures collected, a data dimensionality reduction from all 19 subscales of 7 questionnaires and 2 physical performance scores was achieved using a principal component analysis (PCA) in KOA patients at baseline. This allows us to reduce the data into fewer dimensions, while retaining its trends and patterns.” (line: 273-276). 

Reviewer 3: 

The authors evaluated knee and hip OA patients before, 3 and 6 months post-TJR searching for clinical

predictors of pain persistence. It is nice to see that the reliability is reported for radiographic assessment. However, reliability is extremely important for all the clinical applications of all the assessments. It would be a value of including an independent paragraph as part of the Introduction to highlight the topic on reliability (see and cite this recent commentary https://www.nature.com/articles/s41562-019-0655-x). Especially, the whole project included brain imaging with multiple modalities such as T1, rs-fMRI and DTI although not reported in the present work. On the other hand, the reliability of other assessments reported in the present manuscript should also be presented in the manuscript.

Response: We appreciate the Reviewers opinion on our paper. 

Regarding the reliability for applied measurements, we agree this is an important topic. The radiographic assessment is dependent on the operator; we choose to have 2 observers scoring the x-rays, and thus we report the interrater reliability. For all the other measures, that are mainly questionnaire assessments we have one measure for patient at each time-point. However, measures of internal consistency and test-retest reliability for each questionnaire used are present in the cited bibliography for each scale. We further report this in the methods: 

“ 2.3.4 Questionnaires – Pain, Mood and Health

Seven questionnaires were administered by a trained clinician, during face-to-face interview. They were administered both before surgery (V1), and in the post-surgical visits (V3-V4). The repeated use of the same measures allowed us to track changes concerning intensity and quality of pain, emotion and affect, health and quality of life. All questionnaires were used in their Portuguese version and validation data regarding their adequate context validity, internal consistency and test-retest reliability was consulted and hereby cited. We assessed: 1) KOOS, HOOS, validated injury and OA outcome scores for knee and hip [20-22]; 2) Brief Pain Inventory – Short Form (BPI) [23-25]; 3) McGill Pain Questionnaire (MPQ) [26, 27]; 4) Doleur Neuropathique en 4 Questions (DN4) [25, 28]; 5) Hospital Anxiety and Depression Scale (HADS) [29, 30]; 6) Pain Catastrophizing Scale (PCS) [25, 31]; and 7) SF36-item Short Form Survey (SF36) [32, 33] .”

We believe the topic of sample size and reliability is of particular importance for this paper and as suggested by the Reviewer we add a relevant paragraph and cite the suggested paper (Line 850-869). 

“ The topic of reliability and sample size was recently discussed [43] in the field of neuroimaging but rendering important implications to other areas. The authors point that while sample size is recognized as a major determinant of statistical power, measurements of reliability are less commonly considered, that place an upper limit on the maximum detectable effect size. We measured inter-reliability regarding the radiographic assessment and used questionnaires with high context validity, internal consistency and test-retest reliability. Although we did not measure reliability directly, we believe this is an important remark, and data quality assessments are always important.”

1. The relationship among sample size, measurement reliability, statistical power and effect size has been recently demonstrated in a publication (see https://www.nature.com/articles/s41562-019-0655-x). It is thus critical to report the reliability of all the measurements employed in the manuscript. It would be perfect if the authors can test the reliability directly, otherwise the citation on the previous reliability publication is fine.

Response: This question is addressed above.

2. So many statistical tests are performed, which should be corrected in some forms for multiple comparisons. For examples, FDR methods.

Response: This is a good point. Although we performed multiple tests (4 outcome variables; baseline and post-surgery; network analysis), the analysis were mainly independent from each other. Moreover, this was an exploratory analysis, and the goal when using four distinct pain metrics was to further characterize its relationship with distinct pain related dimensions. As for the network analysis, permutation-based tests are exact and obliviate the need for corrections. 

We believe, given the theoretical frame, hypothesis tested and interpretation of the results, there is no requirement to correct for multiple comparisons. 

3. It is a problem that many tests were done with such a small sample size, which limited the generalizability of the current findings. This must be discussed as a limitation.

Response: We appreciate the Reviewers assessment; the point of generalizability is further commented in the discussion.

---

## [Decision Letter · Decision Letter 1]

16 Dec 2019

Prognostics for pain in osteoarthritis: Do clinical measures predict pain after total joint replacement?

PONE-D-19-23720R1

Dear Dr. Apkarian,

We are pleased to inform you that your manuscript has been judged scientifically suitable for publication and will be formally accepted for publication once it complies with all outstanding technical requirements.

With kind regards,

Yuanyuan Wang

Academic Editor

PLOS ONE

Additional Editor Comments (optional):

The authors have addressed the reviewers' comments properly.

Reviewers' comments:

Reviewer's Responses to Questions

**Comments to the Author**

1. If the authors have adequately addressed your comments raised in a previous round of review and you feel that this manuscript is now acceptable for publication, you may indicate that here to bypass the “Comments to the Author” section, enter your conflict of interest statement in the “Confidential to Editor” section, and submit your "Accept" recommendation.

Reviewer #1: All comments have been addressed

Reviewer #3: All comments have been addressed

2. Is the manuscript technically sound, and do the data support the conclusions?

Reviewer #1: Yes

Reviewer #3: Yes

3. Has the statistical analysis been performed appropriately and rigorously? 

Reviewer #1: Yes

Reviewer #3: Yes

4. Have the authors made all data underlying the findings in their manuscript fully available?

Reviewer #1: Yes

Reviewer #3: Yes

5. Is the manuscript presented in an intelligible fashion and written in standard English?

Reviewer #1: Yes

Reviewer #3: Yes

6. Review Comments to the Author

Reviewer #1: The authors have successfully addressed my concerns, including addressed the issue of small sample size and explained the new statistical models in supporting the conclusion. The results are credible and I have no further comment.

Reviewer #3: The measurement reliability has been discussed in the revision. Especially, the authors claimed the limitation of sample size. I would love to recommend an acceptance of the current manuscript in publication.

7. PLOS authors have the option to publish the peer review history of their article (what does this mean?). If published, this will include your full peer review and any attached files.

Reviewer #1: Yes: Ying Ying Leung

Reviewer #3: No

---

## [Editor Report · Acceptance letter]

26 Dec 2019

PONE-D-19-23720R1 

Prognostics for pain in osteoarthritis: Do clinical measures predict pain after total joint replacement? 

Dear Dr. Apkarian:

I am pleased to inform you that your manuscript has been deemed suitable for publication in PLOS ONE. Congratulations! Your manuscript is now with our production department. 

With kind regards,

on behalf of

Dr. Yuanyuan Wang 

Academic Editor

PLOS ONE